# Alkalinity Regulation and Optimization of Cementitious Materials Used in Ecological Porous Concrete

**DOI:** 10.3390/ma17081918

**Published:** 2024-04-21

**Authors:** Sijiao Li, Jian Yin, Wenxing Xu, Sizhe Liu, Xiaofei Liu

**Affiliations:** School of Civil Engineering, Central South University of Forestry & Technology, Changsha 410004, China; 20210100054@csuft.edu.cn (S.L.); 20211100403@csuft.edu.cn (W.X.); 20221100432@csuft.edu.cn (S.L.); 20221100430@csuft.edu.cn (X.L.)

**Keywords:** ecological porous concrete, alkalinity regulation, chemical admixtures, diatomaceous earth, micro-mechanisms

## Abstract

Ecological porous concrete (EPC) is one of the novel formulations of concrete with unique phytogenic properties. However, achieving both low alkalinity and high strength in EPC proves challenging due to the inherently high alkalinity of the pore environment, which hinders the growth of the plant and affects its ecological benefits significantly. This research investigated the utilization of 15 types of chemical admixtures and diatomaceous earth as alkali-reducing agents to optimize the properties of silicate cementitious materials for the applications of EPC. To identify the most effective agents, the pH value and compressive strength of the cement paste were adopted as the screening criteria for the selection of the essential alkali-reducing ingredients. Subsequently, a composite approach combining chemical admixtures and DE was employed to explore the synergistic effects on the pH and strength of silicate cementitious materials. The results revealed that a combination of 8% DE, 5% oxalic acid, and 5% iron sulfate functioned effectively and resulted in desirable performance for the concrete. This synergistic blend effectively consumed a large amount of Ca(OH)_2_, reducing the pH of cement paste to 10.48 within 3 days. Furthermore, the hydration reaction generated C-S-H with a low Ca/Si ratio, leading to a remarkable increase in the compressive strength of the concrete, reaching 89.7 MPa after 56 days. This composite approach ensured both low alkalinity and high strength in silicate cementitious materials, providing a theoretical basis for the application and promotion of EPC in the ecological field.

## 1. Introduction

Ecological porous concrete (EPC) is a novel form of a randomly porous concrete construction material that can establish flora symbiosis. Its opening and connecting porosity are generally adjusted in the range of 20–30%. It has several functions, including soil and water conservation, plant restoration, water purification, acoustic noise absorption, and structural protection. It has been applied extensively in projects involving parking lots, roof greening, river bank berms, highway berms, and building wall and river bank berms [1,2]. Simultaneously, in sponge city construction, the promotion and application of EPC have great practical significance in repairing urban water ecology, nourishing water resources, improving urban flood prevention, improving the quality of new urbanization, mitigating the urban heat island effect, and promoting human–nature harmony.

However, the cementitious material used to create EPC is largely ordinary silicate cement (OPC), and the highly alkaline pore environment (pH as high as about 13) inside EPC can inhibit normal plant growth (pH between 6 and 9). Therefore, it is of great theoretical significance and engineering application value to select key materials to reduce the alkalinity of silicate cementitious materials and to explore the interrelationship between alkalinity reduction and the mechanical properties of silicate cementitious materials in order to ensure that the EPC can meet the alkaline environment requirements for plant growth and the continuous stability of its mechanical properties. At present, there are five main alkalinity-reduction procedures typically employed in EPC: utilizing low-alkali cement to replace OPC, using mineral admixtures to replace OPC in equal proportions, adding chemical admixtures, spraying or soaking chemical solutions, and employing coatings to seal the alkalinity.

Currently, research on cementitious materials for EPCs focuses on sulfoaluminate cement (SAC), aluminate cement (CA), magnesium ammonium phosphate cement (MPC), and other low-alkalinity cementitious materials. The hydration rate of SAC is relatively fast, and the hydration products are predominantly calcite, monosulfoaluminate, and amorphous alumina hydroxide (AH3) [3]. Therefore, the pore solution pH of EPCs made with SAC is lower than that of OPC. The pH of EPCs prepared by Gong et al. after using urea-modified SAC was lower than 8.31 at 28 d [4]; Gong et al. [5] used 20% limestone powder in an equal amount to replace SAC, and the pH of the EPCs decreased to 9.67; Li et al. [6] added natural zeolite to modify SAC, and the cement paste pore alkalinity was reduced to 10.3; Li et al. [7] modified SAC by using 10% bentonite, and the pH of the cement paste pore solution was reduced to 10.95; Gong et al. [4] prepared EPC using CA, which had an alkalinity increase of 0.2 to 0.4 units compared with SAC, but an increase in the compressive strength of 1 MPa to 1.5 MPa; MPC has also been applied to reduce the alkalinity of EPC—Zhao et al. [8] prepared EPC with MPC, and the pH value was between 6.8 and 8.5, which was much lower than that of OPC; Wu et al. [9] used LACM with autogenous gangue aggregate to prepare EPC, and the pH value at 28 d was reduced to 8.75; in order to study the neutral cementitious materials used for EPC, Lee et al. [10] investigated cementitious materials based on hemihydrate calcium sulfate, and the results showed that, no matter how low the cement content was, it could not reduce the alkalinity of concrete, but the pH of their homemade cementitious material was neutral.

Mineral admixtures, such as silica fume (SF) and fly ash (FA), have volcanic ash activity, which can react with Ca(OH)_2_ in the hydration products of cement and consume some of the OH^−^, thus decreasing the pH. Li et al. [11] employed 10% SF to replace the OPC, and the pH of the cement paste was reduced to 11.0 at 56 d, which represents a reduction of 13.4% compared with the BG. The pH-lowering of fly ash, slag, and acid-modified cement pastes was not considerable compared with silica fume. Li et al. [12] reported a gradual decrease in the 28 d strength of EPC with decreasing pH when they increased the FA replacement from 0% to 30%, but they could not reach the pH required for a plant-growth environment, and a combination of other alkali-reduction techniques was needed to achieve the goal. Kong et al. [13] used three alkali-reducing strategies to reduce the alkalinity of EPC, and the optimum plant growth performance was achieved when the fly ash admixture was 10%. Cheng et al. [14] raised the FA or SF concentration gradually, the alkalinity of the concrete was greatly reduced, and the alkalinity reduction of silica fume was better than that of fly ash. The pH of EPC was decreased to 9.4 by combining 20% fly ash and 10% silica fume.

The addition of chemical admixtures, immersion in chemical solutions, and the application of coatings to seal out alkalis are all chemical alkali-reduction procedures. The addition of acidic and salt chemicals and spraying or immersion in acidic chemical solutions are based on the theory of acid–base neutralization for alkali reduction. Adding polymers and spraying water-repellent chemical solvents can fill the capillary pores inside the cementitious materials and generate a hydrophobic coating on the surface of the cement to restrict the release of alkalis. Kong et al. [13] added 0.4% ferrous sulfate to EPC and 3% potassium dihydrogen phosphate to the soaking solution, which effectively reduced the alkalinity by compound alkali-reducing treatments; Yang et al. [15] used clear water, oxalic acid solution, and ferrous sulfate solution for spraying and soaking EPC, and the pH could be reduced by 0.08–0.53 units after soaking in water, 0.25–0.53 units after soaking in ferrous sulfate, and 0.31–0.57 units after soaking in oxalic acid; Yang et al. [16] used a single-alkali reduction method for EPC and a hybrid alkali-reduction method for EPC, the pH of 28 d was reduced to 10.4 and 10.0, respectively.

A single alkali-reduction procedure utilizing low-alkali cementitious materials or mineral admixtures is not as successful and results in a slight increase in strength. While the chemical alkali-reduction method may efficiently lower the alkalinity inside the EPC, the adverse effect on the strength of the EPC is larger. In comparison, the composite alkalinity-reduction approach is preferable to the single alkalinity-reduction method. Therefore, the technique of composite modification of cementitious materials using chemical admixtures and mineral dopes can be one of the most effective approaches to reducing the alkalinity of EPC.

Based on the principle of botany, the ideal soil pH for plant growth varies according to plant species; most plants grow normally in a soil environment with a pH value of 5.0–9.0, while alkali-tolerant plants can grow normally in a soil environment with a pH value of 9.0–11.0. Initially, the pH value of the cement paste should be less than 11.0 at 3 d and less than 8.6 at 56 d. The loss of compressive strength of the cement paste should not be more than 15% at all ages, and the effect should be measured jointly with pH value and strength.

The purpose of this study is to modify ordinary silicate cementitious materials. Fifteen various chemical admixtures and diatomaceous earth were employed as alkali-reducing materials to pretreat cementitious materials for one-factor alkali-reduction. Alkalinity matching and strength stability of alkali-reducing materials and cementitious materials were researched, and efficient alkali-reducing materials were selected. Secondly, the chemical admixture and diatomaceous earth were used to conduct a compound alkali-reducing test to rationally regulate and reveal the influence mechanism of the optimal compound mixing combination on the pH value and compressive strength of the cementitious materials.

## 2. Material and Experimental

### 2.1. Materials

In this study, ordinary silicate cement with a grade of P.O 42.5, produced by Hunan Pingtang Southern Cement Co. in Changsha, Hunan province, China, was used. It features a specific surface area of 327 m^2^/kg, 3-day compressive and flexural strengths of 28.1 MPa and 5.3 MPa, and 28-day compressive and flexural strengths of 51.8 MPa and 8.06 MPa, respectively.

Adding acids is a typical approach to neutralizing alkaline solutions, and salts containing alkaline metal ions can also be neutralized with bases to create water and salts. In an attempt to reduce the alkalinity of concrete, 15 alkali-reduction materials were considered in this study, which included oxalic acid (OA), silicic acid (SA), salicylate (SAA), ferrous sulfate (FS), ferrous sulfate (FS2), aluminum sulfate (ALS), potassium sulfate (PS), ammonium sulfate (AS), ammonium carbonate (AC), ammonium bicarbonate (AB), potassium dihydrogen phosphate (MP), magnesium silicate (MS), potassium acetate (PA), aluminum chloride (ALC), and calcium superphosphate (CS).

Diatomaceous earth (DE) is a naturally occurring mineral material composed primarily of dead and deposited diatom remains. Diatomaceous earth has a fine particle structure and a large specific surface area. Alkaline cement paste reacts with the siloxane groups on the surface of diatomaceous earth to form substances such as calcium silicate. This improves the performance of concrete while reducing the environmental impact and promoting sustainable development. In this paper, 200 mesh calcined diatomaceous earth produced by Hebei Run Huabang New Material Science and Technology Co. in Shijiazhuang, Hebei province, China, was used. Table 1 lists the general physico-chemical properties of 200 mesh DE.

Polycarboxylate superplasticizer from Sika Ltd. was used as a water-reducing agent, with a water reduction of 20%.

### 2.2. Test Methods

As the components of cementitious materials for preparing EPC are special, they are mainly composed of water, cement, and additives. Therefore, except for the material and proportion of the specimen, the preparation, maintenance and testing of the specimen were carried out in full accordance with the requirements of GB/T 17671-2021 “Test Method of Cement Mortar Strength (ISO method)” [17]. Prismatic samples (40 × 40 × 160 mm^3^) were prepared for testing. After demolding, the samples were maintained at room temperature for 24 h. Subsequently, they were placed in a standard curing box at specific intervals (3, 7, 28, and 56 days) for measuring both the pH value and compressive strength at various curing ages.

To differentiate between test combinations, a unique numbering system was adopted. It consisted of the admixture percentage followed by the abbreviated name of the chemical admixture. For instance, pure cement paste with 3% potassium sulfate was labeled as 3% PS. Diatomaceous earth (DE) was used in other variations, and their designations followed the formula: water–cement ratio + admixture percentage + DE. The mixing ratios and properties of this base group paste (BG) are detailed in Table 2, and all experiments were carried out in accordance with the specifications outlined in it.

#### 2.2.1. pH Value

In this test, three sets of parallel tests were carried out on different types of samples of different ages. The pH value of the net cement paste was tested using the aqueous ambient alkalinity release method. This method requires the samples to be immersed in 500 mL of aqueous solution for 3 d, 7 d, 28 d, and 56 d, respectively, and then the containers were sealed. Preventing contamination of the solution prior to testing ensured the accuracy of the data. After 24 h, the pH value of the solution was measured using a PH818 pH meter manufactured by Sigma Instruments and the data were read when the reading no longer changed. The solution was mixed thoroughly before each measurement. In each of the three parallel sets of tests, three pH readings were taken and the average value of the test was recorded. If the error of one data point was more or less than 10% of the other two, the data point was invalid. Finally, the average of the pH values from the three parallel sets of tests was taken as the final data for the sample.

#### 2.2.2. Compressive Strength

The tests were conducted in accordance with GB/T 17671-2021, “Test Method of Cement Mortar Strength (ISO method)” [17], utilizing a DTE-3008 Microcomputer Servo Cement Flexural and Compression Tester from Wuxi Xinjian Instrument Technology Co. in Wuxi, Jiangsu province, China. This widely recognized standard ensures consistent testing procedures, and the chosen equipment met its requirements, guaranteeing reliable and reproducible results. In this test, the average of the six compressive strength test values obtained for a set of three prisms was used as the experimental result. If only one of the six tests exceeded ±10% of the average of the six, that result was rejected. The average of the remaining five was used as the test result. If any of the five test results exceeded ±10% of the mean value, this set of results was invalidated.

#### 2.2.3. Microscopic Test

The microstructure of the samples was investigated using a TESCAN MIRA LMS Xplore Scanning Electron Microscope (SEM) to understand the micromorphology of the materials.

### 2.3. Study Design

The flow chart of the research in this paper is shown in Figure 1.

## 3. Results

### 3.1. pH and Compressive Strength of Cement Paste with Chemical Admixtures Alone

#### 3.1.1. Acids

Oxalic acid (OA), salicylic acid (SAA), and silicic acid (SA) can liberate hydrogen ions (H^+^) in water and react with Ca(OH)_2_ in cement. The pH and compressive strength of cement pastes doped with different acid chemical admixtures at different ages are presented in Table 3, and scatter plots of the pH and compressive strength are shown in Figure 2. After doping acid chemical admixtures, the pH of cement paste decreased from 10.7 to 11.7 to 8.44 to 8.68 in 3 d–56 d, and the compressive strength increased from 38.73–45.8 MPa to 70.17–87.47 MPa.

The pH of the cement paste all fell to varying degrees with the addition of acid-based chemical admixtures. At 3 d, the pH of the cement paste decreased dramatically when the SA dosage increased from 1% to 5%. The 5% SA admixture demonstrated the best alkalinity reduction with a pH of 10.7, which was 9.2% lower than that of BG, and was projected to boost the plant seed germination rate. At 28 d, when the OA dosage increased from 1% to 5%, the pH values of cement paste were all below 9, which was much lower than those of the other two types of admixtures. At 56 d, the 5% SAA admixture had the best effect, with the pH decreasing to 8.44. The pH of the cement paste was below 8.7 after the addition of these three types of acid-based chemical admixtures, which may satisfy the environmental conditions for the proper growth of most plants. In terms of strength changes, the compressive strength of the majority of the cement pastes steadily decreased with the increase in chemical admixture dosage. At 3 d, 1% OA exhibited the maximum compressive strength of 45.8 Mpa, which was 5% greater than that of BG. At 56 d, 3% SAA had the maximum compressive strength of 87.47 Mpa, which was 11% greater than that of BG. The 5% SA admixture also had a high compressive strength of 80.78 Mpa. The various chemical admixtures with different dosages drastically lowered the strength of the cement paste.

In summary, acid-based chemical admixtures considerably decreased the pH of cement pastes, while leading to a significant decrease in strength. This is because acid reacts with alkali, but also corrodes cement and its hydration products [18,19,20], lowers the surface tension of cement particles, reducing the force of interaction between the particles, alters the cement’s fluidity, and slows down the cement’s setting speed [21]. The early alkali-reducing impact of 5% SA was ideal, whereas the later alkali-lowering effect was rather modest. The later alkali-reducing effect of 5% OA was significant, although the strength loss was roughly 9%. At 56 d, the strength of 3% SAA was the best, and the early pH was greater, although the late pH was in compliance with the requirements. Therefore, 5% OA and 5% SA had excellent alkali-reduction effects, and 3% SAA had an outstanding strengthening impact.

#### 3.1.2. Sulfates

Sulfate has strong solubility and can react with Ca(OH)_2_ in cement. The pH and compressive strength of the cement paste doped with five sulfate-based chemical admixtures at various ages are presented in Table 4, and the pH scatter plot and compressive strength scatter plot are shown in Figure 3.

It can be observed from the graphs that the pH of the cement paste doped with the five sulfate-based chemical admixtures decreased from 11.02–12.0 to 8.37–8.73 and the compressive strength increased from 40.67–49.58 Mpa to 63.73–84.67 Mpa in the range of 3 d–56 d.

At 3 d, 5% AS had the lowest pH of 11.02, which was 6.5% lower than that of BG. At 7 d, PS had the best pre-alkalinity decrease, with 3% PS having a low pH of 9.8, which was a 10.5% reduction, which was good for the pre-growth of plant seeds. At 28 d, 1% PS had the lowest pH of 8.61, a drop of 10.3%. At 56 d, FS2 considerably lowered the pH of the cement paste to 8.37 for 1% FS2, a decrease of 4.3%. In terms of strength changes, FS, FS2, and PS had early strength qualities. At 3 d, 1% PS exhibited the maximum compressive strength of 49.58 Mpa, which was 13.6% greater than that of BG. At 7 d, 3% FS2 exhibited the maximum compressive strength of 67.2 Mpa, a 15% improvement. By 28 d, the 5% FS admixture exhibited the strongest strength development, which increased to 86.57 Mpa, a 9.1% improvement. At 56 d, all sulfate-based chemical admixtures, except for FS2, affected the strength of the cement paste to varying degrees, and the 1% FS2 had the maximum compressive strength of 84.67 Mpa, an enhancement of 7.4%.

In summary, the 1% PS admixture had the most excellent alkali-reduction effect, but the strength damage was 7%. The 5% FS and 1% FS2 admixtures greatly boosted the compressive strength of the cement paste, but the pH value in the early stage could not easily meet the requirements of normal plant growth. The addition of sulfate expedited the hydration process of cement, the amount of hydration products increased, and the pH value was enhanced. Compared with acid-based chemical admixtures, sulfates contributed more to the compressive strength of cement paste, contrary to the findings of this study [22].

#### 3.1.3. Carbonates

Ammonium carbonate (AC) and ammonium bicarbonate (AB), as water-soluble compounds, can react with CaO in cement to create CaCO_3_, NH_3_, and H_2_O. The pH and compressive strength of cement paste combined with two kinds of carbonate chemical additives at various ages are presented in Table 5, and the scatter diagrams of pH and compressive strength are shown in Figure 4. The pH of the cement paste increased from 11.02 to 11.9 to 8.58 to 8.71 in 3 d –56 d and compressive strength increased from 34.45 to 42.2 to 58.93 to 73.9 Mpa.

The alkali-reduction effect of AC was low in the early stage, and the 1% AC exhibited a slight alkali-reduction effect. At 28 d, the pH value of cement paste declined at a higher pace, and the pH values of 1% AC were 8.72 and 8.67 after 28 d and 56 d, respectively. However, the inclusion of AC led to a substantial loss of the compressive strength of cement paste, and at 56 d, the strength was reduced by 15.1% compared with that of BG, which made it impossible to meet the required mechanical properties. Compared with AC, AB had a more visible effect of decreasing the alkalinity in the early stage, but by 28 d, the pH of the cement paste was over 9.0, which does not meet the requirements of the vegetative environment. The inclusion of AB led to a more substantial loss of strength of the cement paste, and at 56 d, the compressive strength of 1% AB was reduced by 25.2%.

In summary, carbonate chemical admixtures have weak alkali-reduction action and more serious strength loss. This is due to the fact that carbonates dissolve in water and react with Ca(OH)_2_ and C-S-H gel in cementitious materials to form insoluble calcium carbonate, which results in a decrease in the pH of the pore solution of the cementitious materials, as well as the deterioration of the pore structure [23,24], thus affecting the strength of the cement.

#### 3.1.4. Other Classes

From 3 d–56 d, the pH value of cement paste fell from 11.1–11.9 to 8.49–8.88, and the compressive strength rose from 40.68–49.53 MPa to 64.5–78.25 Mpa after combining five kinds of various kinds of chemical admixtures. The proportion, pH value, and compressive strength of cement pastes combined with other kinds of chemical admixtures are presented in Table 6, and the scatter plots of the pH value and the compressive strength are also shown in Figure 5.

At 3 d, the pH values of cement pastes with five different chemical admixtures were over 11.0. At 7 d, the pH values of the cement pastes were all above 10.0. At 28 d and 56 d, ALC showed the best alkalinity reduction, with the pH of 5% ALC reducing to 8.47 and 8.49, respectively. This was attributed to the fact that chloride and aluminum ions, through early exothermicity, decreased the setting time by producing calcite, which not only consumed alkalis, but also thus increased the early strength [25,26]. In terms of strength changes, MP and ALC increased the early strength of cement paste. Meanwhile, MS, PA, and CS resulted in varying degrees of losses in the strength of the cement paste. At 28 d, except for 3% MP and 1% CS, the strength losses of the cement pastes in other combinations ranged from 4.5% to 21.9%. At 56 d, the growth rate of the compressive strength of the cement pastes continued to decline. The 5% ALC admixture had the lowest strength of 64.5 MPa, which was 18% lower than that of BG. Compared with the acid and sulfate chemical admixtures, other forms of chemical admixtures exhibited inferior impacts on alkali reduction and strength stability.

By assessing the pH and compressive strength, at 3 d, 5% SA has the best alkali-reduction effect, but created a considerable loss of strength in the latter stage. At 7 d, 3% OA had the best alkali-reducing effect, but the pH of the cement paste at 3 d was 11.70, which did not satisfy the early pH requirement and created a considerable loss of strength at the latter stage. At 56 d, the compressive strengths of 3% SAA and 1% FS2 continued to rise gradually, but their pH values at 3 d were 11.6 and 11.8, respectively, which could not meet the early alkalinity requirement. Therefore, it is difficult for the alkalinity-reduction method with a single chemical additive to reduce the pH of cement paste while retaining the stability of the strength. However, different materials had exceptional impacts on single indexes, and the chemical admixtures and dosages with great alkalinity reduction were 5% OA, 5% SA, and 3% PS, and the chemical admixtures and dosages with excellent compressive strength development were 1% FS2, 5% FS, and 3% SAA.

### 3.2. pH and Compressive Strength of Cement Paste with Single DE

DE is a structural system based on silica, which has a naturally porous structure and strong adsorption capability [27,28]. The fit ratio, pH, and compressive strength of cement pastes doped with DE are presented in Table 7, and the scatter plots of the pH and compressive strength are shown in Figure 6. When the water–cement ratio was increased from 0.25 to 0.35, the pH of the cement pastes with different DE doping quantities decreased from 11.43 to 12.0 to 8.47 to 9.21, and the compressive strength increased from 28.55 to 65.13 Mpa to 68.51 to 85.95 Mpa.

When the water–cement ratio was 0.25, at 3 d, the pH of the cement paste tended to decrease with the increase in DE dosage. At 3 d and 7 d, the pH of 8% DE was the lowest at 11.56 and 9.58, which were 3.7% and 6.1% lower than those of BG, respectively. During the hydration process from 3 d to 7 d, the pH of the cement paste decreased dramatically by 15% to 19%. At 28 d and 56 d, the alkali-reduction impacts of 2% DE were ideal, with pH values reaching 8.48 and 8.47, respectively. According to the strength change, at 3 d, the early compressive strength of 2% DE had a minor increase of 65.13 MPa, which was 4.2%. The compressive strength of 8% DE was the lowest, which was 54.6 MPa, representing a loss of 12.6%. From 3 d to 7 d, the strength of the cement paste developed faster, and the compressive strength of 8% DE increased by 32%. At 56 d, the compressive strengths of 6% DE and 8% DE were 85.95 MPa and 85.57 MPa, respectively, which were 2% and 1.6% higher than that of BG, and the strength development was constant. The cement pastes at other DE doses exhibited reductions in strength. Therefore, considering the balance between pH and compressive strength, the best dosing of DE was 8% at a 0.25 water–cement ratio.

At a water–cement ratio of 0.3, at 3 d, the pH of 4% DE was the lowest, at 11.52, which was 2.3% lower than that of BG. By 7 d, the pH of 6% DE was the lowest, decreasing from 12.0 to 9.74 by 11.1%. At 28 d and 56 d, the pH of 6% DE was the lowest, at 8.68 and 8.62, respectively. In accordance with the change in strength, the compressive strengths of 2% DE and 4% DE were enhanced by 1.6% and 1.4% in 3 d; however, the DE dosage of 6%, 8%, and 10% lowered the compressive strength of the slurry by 6%, 8%, and 10%, respectively. The compressive strength dropped by 6.9%, 6.9%, and 12.5%, respectively. At 28 d and 56 d, the compressive strengths of 4% DE were 84.75 MPa and 85.45 MPa, which were 6.8% and 8.4% higher than that of BG, respectively, while the other doses of DE negatively affected the compressive strengths of the cement pastes to varying degrees. Therefore, considering the balance between the pH and compressive strength, the best dosage of DE was 4% at a 0.30 water–cement ratio.

At a 0.35 water–cement ratio, there was a considerable increase in pH as compared with water–cement ratios of 0.25 and 0.3. An alkali-reducing impact of 10% DE was noticeable at 3 d and 7 d, where the pH values were 11.43 and 10.27, respectively. The pH decline from 3 d to 7 d was larger, but at 28 d and 56 d, DE was unable to reduce the pH of the cement paste in a continuous manner. Based on the strength variations, at 3 d, the compressive strength of the cement paste fell by 0.6% to 17.1% compared with BG as the DE dosage rose. At 28 d, the compressive strength of 2% DE was the highest, at 71.62 MPa, which increased by 5.5%, while the compressive strength of the cement pastes at other doses declined significantly. The greatest compressive strength of 72.69 MPa was recorded for 8% DE at 56 d. Therefore, considering the balance between pH and compressive strength, the ideal dosage of DE is 8% at a water–cement ratio of 0.35.

In summary, the pH of the cement paste doped with DE at a water–cement ratio of 0.25 was generally lower and much stronger than that at 0.3 and 0.35 water–cement ratios. At 3 d and 7 d, the pH of 8% DE was the lowest. At 56 d, the compressive strength of 8% DE was 85.57 MPa, with stable strength development. This is due to the fact that, at a 0.25 water–cement ratio, there was less water in the cement paste, and the cement was less hydrated in the pre-hydration stage, thereby generating relatively less Ca(OH)_2_ and significantly lower alkalinity in the pre-hydration stage. In the later stage of hydration, the volcanic ash activity of DE stimulated the secondary hydration of the cement system, which not only consumed Ca(OH)_2_ in the system, but also created a C-S-H gel, optimized the interfacial transition zone, and refined the pore structure, enhancing the strength [29]. Therefore, the ideal water–cement ratio was 0.25 and the optimum dosage of DE was 8%.

### 3.3. pH and Compressive Strength of Cement Paste with Compounded Alkali Reduction

According to the experimental results, it was observed that it was difficult to balance the low alkalinity and high strength at all ages with a single-alkali-reduction procedure. In order to develop “low alkalinity and high strength” cementitious materials appropriate for EPC, chemical admixtures and mineral dopants were utilized to lower the alkalinity in a synergistic approach. The compound alkali-reduction method was adopted, the dosage of water-reducing agent was 0.25 kg/m^3^, the water–cement ratio was 0.25, an 8% dosage of DE was used to replace the cement in equal amounts, the chemical admixture with an excellent alkali-reduction effect and strength loss range within 15% was mixed, the chemical admixture with stable strength development was added into the cement, and the test method was the same as that in 1.3. The chemical admixtures with excellent alkali-reduction effects were 5% OA, 5% SA, and 3% PS. The chemical admixtures with stable strength development were 1% FS2, 5% FS, and 3% SAA. The ratio, pH, and compressive strength values of the nine alkali-reduction combinations after two–two compounding are provided in Table 8, and the trends of pH and compressive strength are illustrated in Figure 7.

As can be seen from Figure 7a, the pH of each combination decreased from 10.48 to 11.39 to 8.56 to 8.72 as the age increased. At 3 d, the pH of the cement paste after compounded alkali reduction was substantially lower than that of the BG. In the test with single-chemical admixtures, the pH of 5% SA was the lowest, at 10.7 in 3 d. Meanwhile, in the compounding test, the pH of 8% DE-OA-FS was only 10.48, which was 2% lower than that of 5% SA, and the intrinsic breakthrough in early alkalinity offered more favorable circumstances for plant seed viability. The pH values of 8% DE-OA-FS were 9.68, 8.61, and 8.56 at 7 d, 28 d, and 56 d, respectively, which were 12%, 7.8%, and 6.3% lower than those of BG, with a lower pH than that of other compounded combinations.

As can be seen in Figure 7b, the compressive strength of all combinations increased from 53.63 MPa to 67.73 Mpa to 64.45 Mpa to 91.78 Mpa with the increase in age, the compressive strengths of 8% DE-OA-FS were 67.73 Mpa, 65.98 Mpa, 83.15 Mpa, and 89.7 Mpa at 3 d, 7 d, 28 d, and 56 d, respectively, and the compressive strengths of 8% DE-OA-FS were 67.73 Mpa, 65.98 Mpa, 83.15 Mpa, and 89.7 Mpa at 3 d, 7 d, 28 d, and 56 d, respectively. The compressive strength at 56 d was raised by 8.4%, 3.2%, and 6.5% compared with BG, respectively, whereas the strength at 28 d was decreased by 0.7%, with a more steady strength development. In addition, the later compressive strengths of 8% DE-SA-FS and 8% DE-SA-SAA were higher, and at 56 d, the compressive strengths were 93 Mpa and 91.78 Mpa, respectively, although the compressive strengths were significantly higher than those of the other combinations of single-doped and compound-doped treatments. However, by 28 d, the pH values of 8% DE-SA-FS and 8% DE-SA-SAA were 9.53 and 9.05, respectively, and the late pH values were higher and did not meet the requirements. In the comprehensive analysis, the 8% DE-OA-FS compounded alkali-reducing combination had the best result, which took into account the low pH value of cement at all ages and the stable increase in the compressive strength.

### 3.4. Analysis of Alkali Reduction Mechanism

The microscopic morphology of the samples containing 8% DE-OA-FS is illustrated in Figure 8a,b, which demonstrates the hydration products in the micropores of the samples.

The C-S-H gels were stacked in layers to generate a pore structure. Reticulated and needle-and-rod-shaped chalcocite and calcium hydroxide grew intermingled in the dense C-S-H gel, with a significant number of hydration products.

The major component of DE was amorphous SiO_2_ with small amounts of Al_2_O_3_, Fe_2_O_3_, CaO, MgO, and organic matter, and compared with manufactured SiO_2_, DE had better thermal stability and other chemical properties [30]. In Figure 8c, the DE incorporated into the cementitious material is uniformly distributed, and dense nanopores arranged in an orderly manner on the surface of DE can be observed in Figure 8d. In Figure 8e,f, Aft and C-S-H gels were formed both on the surface and within the nanopores of DE. In the early stage of cement hydration, the addition of DE refined the pore structure of the cement paste, and DE had an adsorption effect on Ca^2+^ and promoted the nucleation, as well as the growth, of C-S-H, which is conducive to the homogeneous distribution and growth of the hydration products [31], thus lowering the pH value of the pore water environment. DE can store a large amount of water in the early stage of hydration because of its porous structure and large specific surface area, and with the continuing progress of the hydration reaction, the cement paste will be able to store a large amount of water at the beginning of hydration. As the hydration reaction continues, the free water content inside the cement paste decreases, and the water contained inside the DE will be released slowly, which supports the further hydration of the cement through the “self-conservation” effect. Secondly, the amorphous SiO_2_ in DE can react with Ca(OH)_2_ in cement to produce denser C-S-H, which consumes Ca(OH)_2_, reduces alkalinity, and improves strength; additionally, the adsorption, microaggregate effect, self-conditioning, and volcanic ash activity of DE can make the structure of cementitious materials more dense. pH and the enormous production of Aft and C-S-H gels improved the compressive strength of the cement paste.

OA and FS also brought considerable changes to the internal system structure and elemental distribution of the cement. The elemental distribution spectra obtained by EDS spectral sweeping of samples with 8% DE-OA-FS and BG are shown in Figure 9.

It can be seen that the contents of the Si and Fe elements in the hydration products increased significantly, but the content of Ca elements decreased with the increase in the type of alkali-reducing materials due to the synergistic effect of DE, OA, and FS and thus caused a significant decrease in the Ca/Si ratio. A particular concentration of OA, as a stimulator of the cement hydration process, functions as an effective complexing agent for the free Ca^2+^ contained in the cement [32]. The drop in the pH and calcium ion concentration of the cement system and the production of calcium oxalate precipitation resulted in a denser structure of the cement paste. The addition of FS, likewise, resulted in the substitution of some of the Ca^2+^ by high-valent Fe^3+^ to form Fe(OH)_3_. The acceleration of the hydration process of the cement lowered the pH inside the cement system, and the SO_4_^2−^ hastened the formation of AFt. The increase in the quantity and quality of hydration products boosted the compressive strength [33]. The synergistic action of DE, OA, and FS resulted in the development of C-S-H [34] gels with reduced Ca/Si ratios during the cement-hydration process, and low-alkali–high-strength cementitious materials for EPC were created.

## 4. Conclusions

In this study, by adding 15 chemical admixtures and DE at different dosages, the effects of single and compound mixing methods on the pH and compressive strength of cementitious materials were discussed, respectively, and the mechanism of action of the effective alkali reduction combinations was analyzed, which led to the following main conclusions:The 5% OA, 5% SA, and 3% PS admixtures are good for reducing the pH of cement paste, and 1% FS2, 5% FS, and 3% SA may considerably enhance the compressive strength of cementitious materials. The ideal water–cement ratio of DE as an alkali-reducing chemical for replacing cement is 0.25, and the optimal dosage is 8%;Compared with BG, 8% DE-OA-FS decreased the pH by 12.7% and 6.3% at 3 d and 56 d, respectively, and enhanced the compressive strength by 8.4% and 6.5%, respectively. The synergistic action of DE, OA, and FS consumed a large amount of Ca(OH)_2_ in the cement paste, which resulted in a decrease in alkalinity and the creation of a C-S-H gel with a low Ca/Si ratio, which considerably increased the compressive strength of the cement paste;The composite alkalinity-reducing combination of DE, OA, and FS balanced the low pH and high compressive strength of cementitious materials, providing a theoretical basis for the application of low-alkali–high-strength cementitious materials in EPC.

## 5. Patents

Jian Yin has applied for a patent entitled “A Low-Alkali High-Strength Cementitious Material for Eco-Porous Concrete and a Preparation Method Thereof”, the current status of which is pending.

## Figures and Tables

**Figure 1 materials-17-01918-f001:**
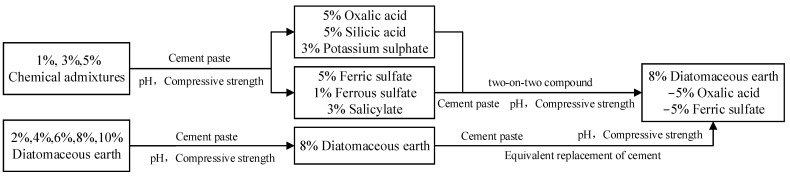
Research flow chart.

**Figure 2 materials-17-01918-f002:**
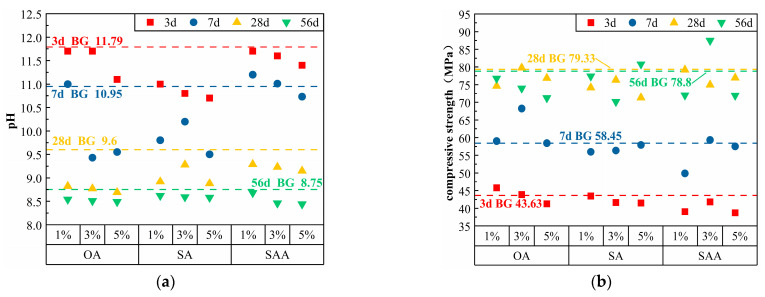
Test results of cement paste after adding acid additives: (**a**) pH; (**b**) compressive strength.

**Figure 3 materials-17-01918-f003:**
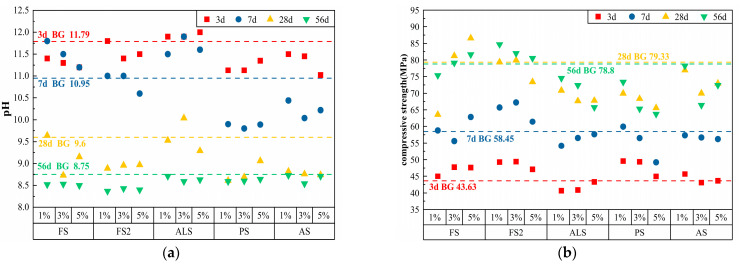
Test results of cement paste after adding sulfate additives: (**a**) pH; (**b**) compressive strength.

**Figure 4 materials-17-01918-f004:**
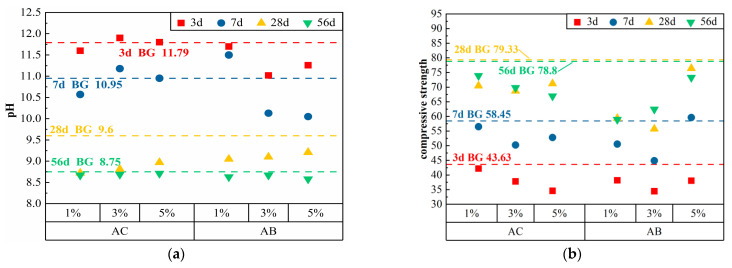
Test results of cement paste after adding carbonate additives: (**a**) pH; (**b**) compressive strength.

**Figure 5 materials-17-01918-f005:**
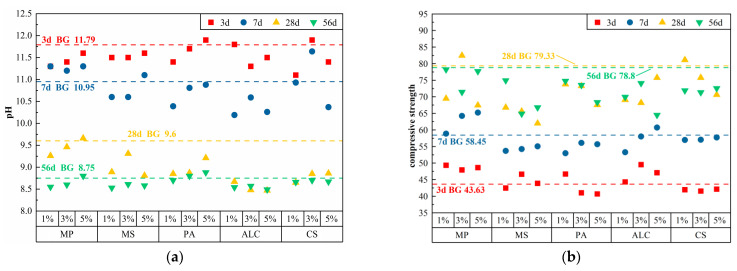
Test results of cement paste after adding other types of chemical admixtures: (**a**) pH; (**b**) compressive strength.

**Figure 6 materials-17-01918-f006:**
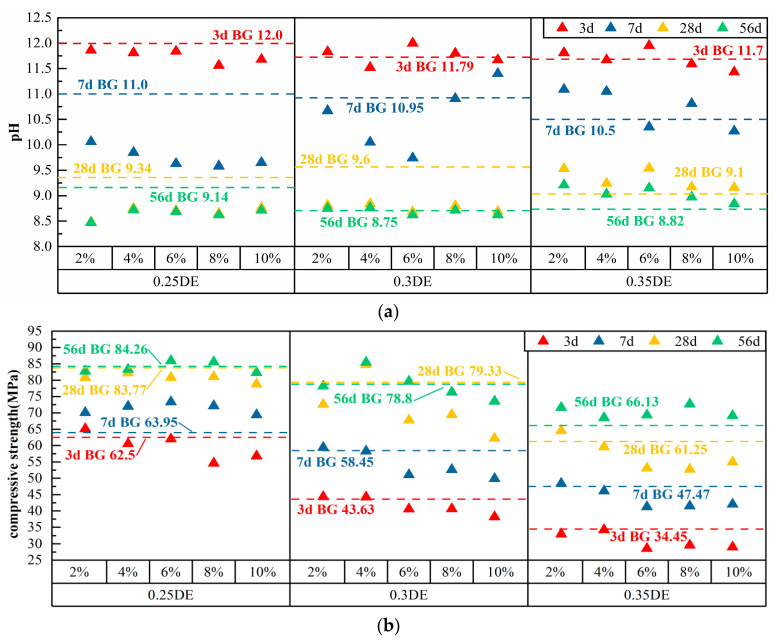
Test results of cement paste after adding DE: (**a**) pH; (**b**) compressive strength.

**Figure 7 materials-17-01918-f007:**
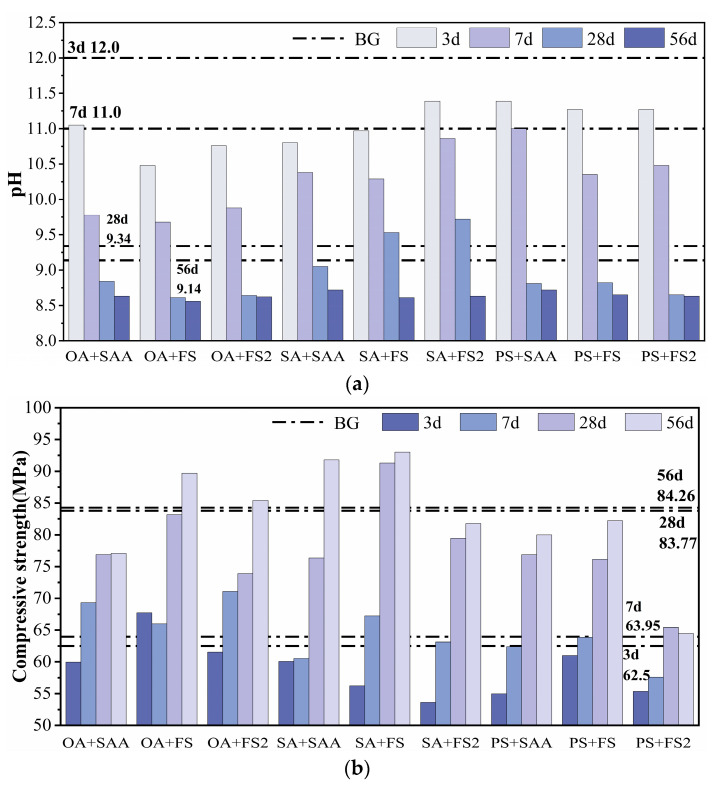
Test results of cement paste with compounded alkali reduction: (**a**) pH; (**b**) compressive strength.

**Figure 8 materials-17-01918-f008:**
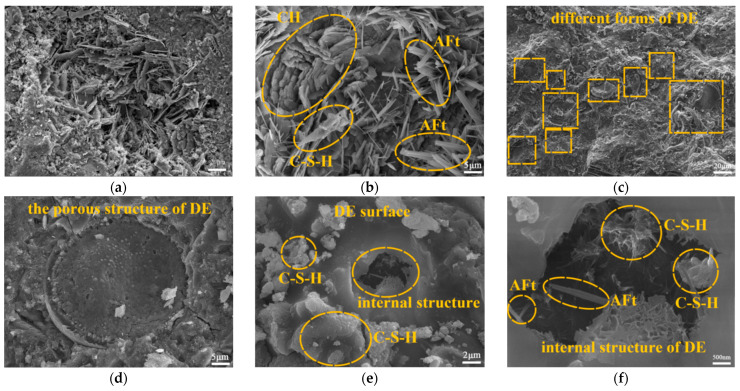
Microscopic morphology of 8% DE-OA-FS doped sample: (**a**) multiple hydration product forms; (**b**) microscopic morphology of hydration products; (**c**) DE and cement fusion form; (**d**) DE; (**e**) DE surface hydration products; (**f**) hydration products.

**Figure 9 materials-17-01918-f009:**
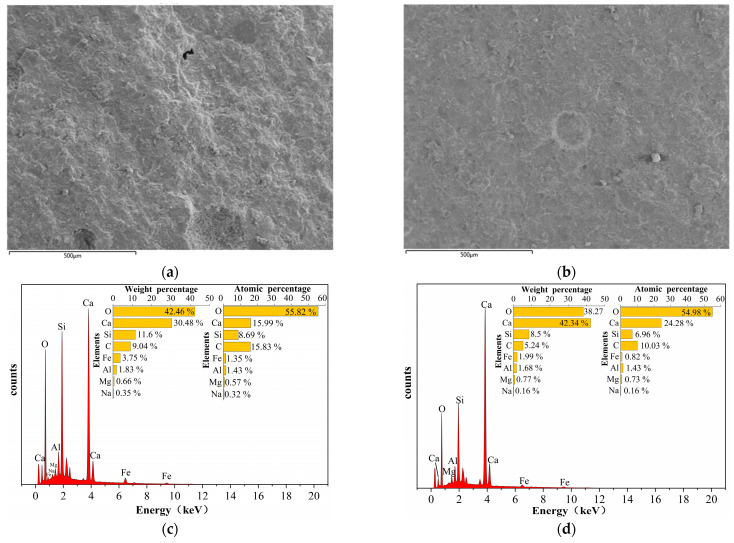
Microscopic morphology of 8% DE-OA-FS-doped sample:(**a**) 8% DE-OA-FS scanning surface; (**b**) BG scanning surface; (**c**) 8% DE-OA-FS elemental distribution spectra; (**d**) BG elemental distribution spectra.

**Table 1 materials-17-01918-t001:** General physico-chemical properties of DE.

SiO_2_ (%)	Fe_2_O_3_ (%)	Al_2_O_3_ (%)	Loss on Ignition (%)	Bulk Density (g/cm^3^)	pH
≥85	≤1.3	≤3.35	≤2.0	0.35–0.39	8–10

**Table 2 materials-17-01918-t002:** Proportion of base cementitious material.

Group	W/C Ratio	Water (kg/m^3^)	Cement (kg/m^3^)	Water Reducer (kg/m^3^)	pH				Compressive Strength (MPa)			
					3 d	7 d	28 d	56 d	3 d	7 d	28 d	56 d
BG	0.25	423	1691	0.25	12.00	11.00	9.34	9.14	62.50	63.95	83.77	84.26
	0.30	507			11.79	10.95	9.60	8.75	43.63	58.45	79.33	78.80
	0.35	592			11.70	10.50	9.10	8.82	34.45	47.47	61.25	66.13

Note: The base group (BG) represents pure cementitious paste without any admixtures.

**Table 3 materials-17-01918-t003:** Mixing ratios and test results for acid-doped cement pastes.

Group		Cement (kg/m^3^)	Water (kg/m^3^)	Water Reducer (kg/m^3^)	Additive (kg/m^3^)	pH				Compressive Strength (MPa)			
						3 d	7 d	28 d	56 d	3 d	7 d	28 d	56 d
OA	1%	1692	507	0.25	5.07	11.70	11.00	8.82	8.54	45.80	59.03	74.63	76.75
	3%	1692	507	0.25	15.21	11.70	9.43	8.77	8.51	43.88	68.25	79.75	73.98
	5%	1692	507	0.25	25.35	11.10	9.55	8.69	8.49	41.28	58.45	76.85	71.23
SA	1%	1692	507	0.25	5.07	11.00	9.80	8.92	8.62	43.48	55.98	74.17	77.37
	3%	1692	507	0.25	15.21	10.80	10.20	9.28	8.59	41.65	56.35	76.33	70.17
	5%	1692	507	0.25	25.35	10.70	9.50	8.88	8.58	41.50	57.95	71.30	80.78
SAA	1%	1692	507	0.25	5.07	11.70	11.20	9.29	8.68	39.05	49.90	79.18	72.00
	3%	1692	507	0.25	15.21	11.60	11.01	9.23	8.46	41.83	59.33	74.95	87.47
	5%	1692	507	0.25	25.35	11.40	10.73	9.15	8.44	38.73	57.55	76.93	71.93

**Table 4 materials-17-01918-t004:** Mixing ratios and test results of sulfate-doped cement pastes.

Group		Cement (kg/m^3^)	Water (kg/m^3^)	Water Reducer (kg/m^3^)	Additive (kg/m^3^)	pH				Compressive Strength (MPa)			
						3 d	7 d	28 d	56 d	3 d	7 d	28 d	56 d
FS	1%	1691	507	0.25	5.07	11.40	11.8	9.64	8.52	45.00	58.77	63.57	75.33
	3%	1691	507	0.25	15.21	11.30	11.50	8.73	8.53	47.75	55.60	81.23	79.10
	5%	1691	507	0.25	25.35	11.20	11.20	9.15	8.50	47.63	62.85	86.57	81.63
FS2	1%	1691	507	0.25	5.07	11.80	11.00	8.89	8.37	49.27	65.70	79.37	84.67
	3%	1691	507	0.25	15.21	11.40	11.00	8.96	8.43	49.40	67.20	79.93	81.97
	5%	1691	507	0.25	25.35	11.50	10.60	8.97	8.40	47.10	61.43	73.40	80.53
ALS	1%	1691	507	0.25	5.07	11.90	11.50	9.53	8.71	40.67	54.17	70.83	74.50
	3%	1691	507	0.25	15.21	11.90	11.90	10.04	8.59	40.88	56.53	67.63	72.38
	5%	1691	507	0.25	25.35	12.00	11.60	9.29	8.63	43.28	57.65	67.78	65.73
PS	1%	1691	507	0.25	5.07	11.13	9.90	8.61	8.59	49.58	59.90	69.95	73.35
	3%	1691	507	0.25	15.21	11.13	9.80	8.69	8.60	49.35	56.50	68.33	65.28
	5%	1691	507	0.25	25.35	11.35	9.89	9.06	8.64	44.93	49.20	65.58	63.73
AS	1%	1691	507	0.25	5.07	11.50	10.44	8.82	8.73	45.68	57.33	76.93	78.25
	3%	1691	507	0.25	15.21	11.45	10.04	8.76	8.54	43.08	56.67	69.98	66.38
	5%	1691	507	0.25	25.35	11.02	10.22	8.74	8.71	43.63	56.20	72.93	72.40

**Table 5 materials-17-01918-t005:** Mixing ratios and test results of carbonate-based cement pastes.

Group		Cement(kg/m^3^)	Water (kg/m^3^)	Water Reducer (kg/m^3^)	Additive (kg/m^3^)	pH				Compressive Strength (Mpa)			
						3 d	7 d	28 d	56 d	3 d	7 d	28 d	56 d
AC	1%	1691	507	0.25	5.07	11.60	10.57	8.72	8.67	42.20	56.50	70.47	73.90
	3%	1691	507	0.25	15.21	11.90	11.18	8.81	8.69	37.80	50.23	68.68	69.80
	5%	1691	507	0.25	25.35	11.80	10.95	8.97	8.71	34.58	52.80	71.20	66.93
AB	1%	1691	507	0.25	5.07	11.70	11.50	9.05	8.63	38.18	50.55	59.33	58.93
	3%	1691	507	0.25	15.21	11.02	10.13	9.10	8.67	34.45	44.88	55.73	62.45
	5%	1691	507	0.25	25.35	11.26	10.05	9.21	8.58	38.05	59.63	76.40	73.28

**Table 6 materials-17-01918-t006:** Mixing ratios and test results of cement pastes mixed with other types of chemical admixtures.

Group		Cement (kg/m^3^)	Water (kg/m^3^)	Water Reducer (kg/m^3^)	Additive (kg/m^3^)	pH				Compressive Strength (Mpa)			
						3 d	7 d	28 d	56 d	3 d	7 d	28 d	56 d
MP	1%	1691	507	0.25	5.07	11.30	11.30	9.26	8.55	49.33	58.88	69.47	78.25
	3%	1691	507	0.25	15.21	11.40	11.20	9.46	8.60	47.93	64.23	82.43	71.43
	5%	1691	507	0.25	25.35	11.60	11.30	9.66	8.80	48.63	65.23	67.43	77.68
MS	1%	1691	507	0.25	5.07	11.50	10.60	8.89	8.53	42.48	53.68	66.78	74.95
	3%	1691	507	0.25	15.21	11.50	10.60	9.31	8.61	46.65	54.23	65.60	64.87
	5%	1691	507	0.25	25.35	11.60	11.10	8.81	8.58	43.88	55.05	61.98	66.77
PA	1%	1691	507	0.25	5.07	11.40	10.39	8.85	8.70	46.68	52.98	73.80	74.77
	3%	1691	507	0.25	15.21	11.70	10.81	8.87	8.80	41.03	56.10	73.27	73.53
	5%	1691	507	0.25	25.35	11.90	10.88	9.21	8.88	40.68	55.70	67.53	68.35
ALC	1%	1691	507	0.25	5.07	11.80	10.19	8.67	8.54	44.30	53.28	69.08	69.93
	3%	1691	507	0.25	15.21	11.30	10.59	8.48	8.57	49.53	58.03	68.18	74.10
	5%	1691	507	0.25	25.35	11.50	10.26	8.47	8.49	47.08	60.73	75.75	64.50
CS	1%	1691	507	0.25	5.07	11.10	10.93	8.65	8.66	41.95	56.98	81.13	71.88
	3%	1691	507	0.25	15.21	11.90	11.64	8.85	8.70	41.55	57.05	75.77	71.33
	5%	1691	507	0.25	25.35	11.40	10.37	8.86	8.67	42.13	57.73	70.60	72.55

**Table 7 materials-17-01918-t007:** Mixing ratios and test results of DE-doped cement paste.

Group		Cement (kg/m^3^)	Water (kg/m^3^)	Water Reducer (kg/m^3^)	Additive (kg/m^3^)	pH				Compressive Strength (Mpa)			
						3 d	7 d	28 d	56 d	3 d	7 d	28 d	56 d
0.25 DE	2%	1658	423	0.25	34	11.86	10.06	8.48	8.47	65.13	70.08	80.83	82.76
	4%	1624	423	0.25	68	11.81	9.85	8.75	8.72	60.55	71.98	82.25	83.28
	6%	1590	423	0.25	101	11.84	9.63	8.71	8.68	62.00	73.35	80.83	85.95
	8%	1556	423	0.25	135	11.56	9.58	8.65	8.62	54.60	72.15	81.07	85.57
	10%	1522	423	0.25	169	11.68	9.65	8.76	8.71	56.78	69.40	78.78	82.31
0.3 DE	2%	1658	507	0.25	34	11.83	10.67	8.81	8.75	44.33	59.40	72.60	78.16
	4%	1624	507	0.25	68	11.52	10.05	8.84	8.76	44.25	58.30	84.75	85.45
	6%	1590	507	0.25	101	12.00	9.74	8.68	8.62	40.60	51.05	67.80	79.65
	8%	1556	507	0.25	135	11.80	10.91	8.80	8.71	40.63	52.65	69.43	76.34
	10%	1522	507	0.25	169	11.67	11.40	8.69	8.62	38.18	49.88	62.20	73.56
0.35 DE	2%	1658	592	0.25	34	11.81	11.09	9.53	9.21	32.95	48.40	64.57	71.62
	4%	1624	592	0.25	68	11.67	11.05	9.24	9.03	34.23	46.08	59.60	68.51
	6%	1590	592	0.25	101	11.95	10.35	9.54	9.15	28.55	41.25	53.08	69.34
	8%	1556	592	0.25	135	11.59	10.81	9.17	8.97	29.6	41.48	52.70	72.69
	10%	1522	592	0.25	169	11.43	10.27	9.16	8.84	29.0	42.05	54.93	69.14

**Table 8 materials-17-01918-t008:** Mixing ratios and test results of DE-doped cement paste.

Group	Water (kg/m^3^)	Cement (kg/m^3^)	DE (kg/m^3^)	Alkali Reducer (kg/m^3^)	Enhancer (kg/m^3^)	pH				Compressive Strength (MPa)			
						3 d	7 d	28 d	56 d	3 d	7 d	28 d	56 d
OA+SAA	422	1556	135	21.14	12.69	11.05	9.78	8.84	8.63	59.95	69.30	76.88	77.08
OA+FS	422	1556	135	21.14	21.14	10.48	9.68	8.61	8.56	67.73	65.98	83.15	89.70
OA+FS2	422	1556	135	21.14	4.23	10.76	9.88	8.64	8.62	61.55	71.10	73.88	85.40
SA+SAA	422	1556	135	21.14	12.69	10.80	10.38	9.05	8.72	60.05	60.53	76.35	91.78
SA+FS	422	1556	135	21.14	21.14	10.97	10.29	9.53	8.61	56.23	67.23	91.28	93.00
SA+FS2	422	1556	135	21.14	4.23	11.39	10.86	9.72	8.63	53.63	63.13	79.43	81.78
PS+SAA	422	1556	135	12.69	12.69	11.39	11.00	8.81	8.72	54.95	62.38	76.87	80.00
PS+FS	422	1556	135	12.69	21.14	11.27	10.35	8.82	8.65	61.00	63.85	76.15	82.20
PS+FS2	422	1556	135	12.69	4.23	11.27	10.48	8.65	8.63	55.35	57.60	65.45	64.45

## Data Availability

Data are contained within the article.

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
