# Peer review of "Alkalinity Regulation and Optimization of Cementitious Materials Used in Ecological Porous Concrete"

_materials, 2024, doi:10.3390/ma17081918_

Round 1

Reviewer 1 Report

Comments and Suggestions for Authors

The manuscript: “Alkalinity Regulation and Optimization of Cementitious Materials Used in EPC” presents the utilization of chemical admixtures and a diatomaceous earth as alkali-reducing agents to optimize the properties of silicate cementitious materials. The results are of interest, and the manuscript is generally well written.

Several points about the article are aimed at improving the paper:

  1. Please check and correct several typing errors throughout the manuscript: e.g. Line 24, “, Providing..”, Line 74, “li et al. … ”, Line 83, insert a space before Cheng …
  2. Title: Please use the full name for Ecological Porous Concrete in place of EPC. This is a not a well-known term for all readers
  3. In my opinion, Fig. 1 should be moved at Material and experimental section, in a subsection named Study design
  4. Material and experimental section  - Please provide the type/model and producers of the used equipment (e.g. pH, SEM, Compression Tester, etc.)
  5. Table 1 – please provide how the properties were measured / established. Please use genera physico-chemical properties in place of foundation properties of DE.
  6. The data for pH and compressive strength presented in Table 2, 3, 4, 5, 6: Please provide the number of measurements for each parameter, and the errors for those measurements (preferable +/- for each parameter or at least a percentage for the measurement uncertainty for pH and compressive strength) .
  7. Please describe how the quality control was assured for the measurements

Author Response

Thank you very much for your suggestion. The author's notes to reviewer are detailed in the uploaded manuscript.

Reviewer 2 Report

Comments and Suggestions for Authors

The paper is dedicated to establishing a correlation between the lowering of the pH value and strength of cement matrix. In some cases of the use of concrete, there is a necessity to decrease pH of a cement stone. However, pH can be decreased to a certain limit, because when pH is lower than 9, the stability of the main phases of cement hydration products CSH sharply goes down, resulting in concrete degradation. For example, under intensive action of CO2, after carbonation of portlandite and decrease of pH below 9, carbonation of CSH can also take place. As a result, CaCO3 and silica gel leading to degradation of the cement matrix can be formed. In the presence of sulfates and carbonates, recrystallization of the CSH-phase can also take place with the transfer into less stable phases like thaumasite and others. All this allows to conclude that the target of the research is actual.

However, an analysis of the submission shows that the target of the research was not reached by the authors:

The durability of a cement matrix in the conditions of pH lowering cannot be evaluated by changes in strength. Other characteristics such as deformability, water resistance, freeze-thaw resistance, and others are also important. Without these characteristics, the conclusions made by the authors are not correct concerning the target of the research.

Additionally, the research was performed with not right methodological approach, namely:

The literature survey does not contain enough information about a problem associated with pH lowering and cement matrix degradation.

Section 2, Table 1. Loss on Ignition and not “heat loss”. What is meant under “hydration”?

Section 2, Table 2. The chosen values of W/C for a cement paste are not substantiated. Per the applicable standard GB-T17671-2021, cement strength is to be determined on a cement mortar (cement: sand) with W/C=0.5. In their research, the authors used a water-reducing agent. In this case, tests of the specimens of various compositions should be performed on a cement paste or cement mortar of equal consistency.

Section 2.2.1. There is no reference to a standard for pH measuring.

Section 3. Test results are produced on cement paste. The tests of the porous concretes are not presented at all. For this reason, the title of the submission is not adequate for the research.

The name of the applicable standard GB-T17671-2021 is referred to incorrectly and differently (lines 153 and 175). Moreover, per this standard, cement strength is to be determined on cement mortar (cement: sand). The authors performed strength using a cement paste.

Author Response

(The authors gave the same response as above.)

Reviewer 3 Report

Comments and Suggestions for Authors

In this paper, the analysis of Ecological Porous Concrete (EPC) is presented in great detail, and the analysis of the potential application of concrete with a low alkali content was done in a high-quality manner. In addition, the mechanical characteristics have even been improved.

As a researcher dealing with similar composite materials, I am interested in whether you have a more detailed explanation of the mechanism of the synergistic effect of DE, OA, and FS on reducing the alkalinity of the cement composite? Is it possible to determine a rule that could be applied to other alkali-activated materials in the future?

Figures 9 a and b: The magnification information is missing in the images or bar line.

References are listed in accordance with the rules of the journal.

The English language is solid.

Also, I strongly support the idea of a patent application, because these materials need to be further developed.

Author Response

(The authors gave the same response as above.)

Round 2

Reviewer 1 Report

Comments and Suggestions for Authors

The authors well responded to the comments. I think the manuscript is acceptable for publication.

Reviewer 2 Report

Comments and Suggestions for Authors

After re-reviewing the manuscript, I have observed that the authors had not adequately addressed all the comments raised during the previous review and provided the answers (at least I have not received the comments of the authors):

The authors did not complete the literature overview with the analysis of the research works related to a relationship between the durability of concrete and the pH of cement stone.

The authors did not complete the submission, as was recommended, by the results obtained on specimens prepared from the porous concrete. The authors also did not accept my proposal to change the title of the submission (I proposed to exclude “porous concrete”). The authors could leave the porous concrete after completing the submission with the results of tests made on porous concrete.

As was already mentioned, the specimens were prepared not following the requirements of the standard GB -T17671-2021 referred to by the authors. According to this standard, the testing should be done using a cement-sand mortar with W/C=0.5. For this reason, the results of testing the mechanical strength of the beam specimens 40x40x160 mm prepared from a cement paste cannot be considered adequate to support the conclusions.